# Characterization of Leukocyte- and Platelet-Rich Plasma Derived from Female Collage Athletes: A Cross-Sectional Cohort Study Focusing on Growth Factor, Inflammatory Cytokines, and Anti-Inflammatory Cytokine Levels

**DOI:** 10.3390/ijms241713592

**Published:** 2023-09-02

**Authors:** Tomoharu Mochizuki, Takashi Ushiki, Katsuya Suzuki, Misato Sato, Hajime Ishiguro, Tatsuya Suwabe, Mutsuaki Edama, Go Omori, Noriaki Yamamoto, Tomoyuki Kawase

**Affiliations:** 1Department of Orthopaedic Surgery, Graduate School of Medical and Dental Sciences, Niigata University, Niigata 951-8510, Japan; tommochi121710@gmail.com; 2Division of Hematology and Oncology, Graduate School of Health Sciences, Niigata University, Niigata 951-9518, Japan; tushiki@med.niigata-u.ac.jp; 3Department of Transfusion Medicine, Cell Therapy and Regenerative Medicine, Niigata University Medical and Dental Hospital, Niigata 951-8520, Japan; katsuyasuzu.xq1@nuh.niigata-u.ac.jp (K.S.);; 4Department of Hematology, Endocrinology and Metabolism, Faculty of Medicine, Niigata University, Niigata 951-8510, Japan; power@med.niigata-u.ac.jp (H.I.); tsuwabe@med.niigata-u.ac.jp (T.S.); 5Department of Health and Sports, Faculty of Health Sciences, Niigata University of Health and Welfare, Niigata 950-3102, Japan; edama@nuhw.ac.jp (M.E.); omori@nuhw.ac.jp (G.O.); 6Department of Orthopaedic Surgery, Niigata Rehabilitation Hospital, Niigata 950-3304, Japan; nirehp.yamamoto@aiko.or.jp; 7Division of Oral Bioengineering, Graduate School of Medical and Dental Sciences, Niigata University, Niigata 951-8514, Japan

**Keywords:** leukocyte- and platelet-rich plasma, female college athletes, PDGF-BB, TGFβ1, IL-1β, IL-1RA

## Abstract

Platelet-rich plasma (PRP) has been increasingly used in sports medicine owing to its various advantages. The purpose of our project was to standardize the parameters before performing large-scale clinical trials in the near future to precisely evaluate individual PRP quality. To examine the effects of regular exercise on PRP quality, this study focused on young female athletes, who have been relatively less studied. Blood samples were obtained from female college athletes (n = 35) and ordinary healthy adults (n = 30), which were considered as controls, and leukocyte-rich PRP (L-PRP) was prepared manually. Body composition indices were determined using a bathroom weight scale equipped with an impedance meter. Growth factors and cytokines were quantified using ELISA kits. Platelet-derived growth factor-BB (PDGF-BB) and Transforming-growth factors β1 (TGFβ1) levels (per platelet) in L-PRP were significantly lower in female athletes than in controls. In contrast, Interleukin-1β and Interleukin 1 receptor antagonist (IL-1RA) levels (per platelet and L-PRP) in L-PRP were significantly higher in athletes, and this difference was more prominent in IL-1RA. These findings suggest that L-PRP from athletes may facilitate the inflammatory phase of the healing process by regulating the pro-inflammatory and anti-inflammatory balance. These chemical compositions can be adopted as “must-check” parameters to characterize individual PRP preparations prior to clinical trials.

## 1. Introduction

Platelet-rich plasma (PRP) therapy is used during the healing process of the body [1,2,3]. PRP injected at the injury site is expected to reproduce this scenario or facilitate the ongoing repair process mainly through growth factors released from platelets (PLTs). Owing to its advantages of safety, surgical invasiveness, injury leave, and cost-effectiveness, PRP therapy is now widely applied in various medical fields, including sports medicine [4,5]. Furthermore, in professional sports, widespread commercial interest can be a major motivation to choose this treatment option [6]. However, as questioned in other medical fields, the effectiveness of PRP in sports medicine remains controversial. PRP treatment has been supported majorly by clinical studies lacking appropriate controls [7] or based on doctors’ personal clinical experience.

In contrast, a limited number of high-quality randomized controlled trials (RCTs) suggested that PRP treatment has no clear benefit in tissue repair in athletes [7,8,9,10,11]. Thus, the number of clinicians who are strongly skeptical about the effectiveness of PRP is increasing [12,13]. Considering the difficulties in the standardization or classification of PRP quality, patient physical condition, injection protocols, and rehabilitation protocols [7,14,15], we believe that such high-quality evidence does not necessarily rule out the possibility of PRP effectiveness.

To address this controversial issue at the preclinical level, it may be helpful to classify patients according to their age, sex, and other indices. Based on the hypothesis that regular intense physical training can affect patients’ sensitivity and responsiveness to PRP treatment and PRP quality, this study focused on regular training levels and investigated the PRP quality of professional male soccer players who received regular intense physical training along with appropriate controls [16,17]. The positive clinical experience that PRP treatment seems to be more effective in athletes than in non-athletic patients was also a motivation for the present study. We demonstrated that growth factor levels in pure PRP (P-PRP) preparations, in which leukocytes were poor in number, were significantly lower in male athletes than in their sedentary counterparts. Interestingly, despite the higher basal metabolic rate (BMR), PLTs’ ATP and polyphosphate levels of PLTs appeared to be lower in male athletes than in their male counterparts.

However, it remains unclear whether this observation can be extended to female athletes. Substantial differences in the physical conditions between male and female athletes are well known. Compared with male athletes, female athletes, particularly in their growth and development stages, have differences in skeletal structures, biomechanical characteristics, and hormone balances [18,19,20,21]. These differences are summarized and represented by the term “in the female athlete triad” (FAT) [22,23,24]. The FAT is used to compare athletes with sedentary females. For example, compared with sedentary females, athletic females with FAT tend to be injured easily, particularly in bone tissue, due to estrogen deficiency, and require more time to recover [22,25].

Because of the difference in basal blood parameters differ between males and females [26,27], PRP composition can be significantly influenced by sex [28,29]. However, there are some conflicting publications indicating an irrelevant influence of sex on PLT concentrations and no significant influence of sex on growth factor levels [30,31,32]. A similar contrast may be observed in the comparison between female athletes and their sedentary counterparts. To the best of our knowledge, to date, no previous study has reported such differences.

The ultimate goal of our ongoing project was to make PRP treatment more predictable and potent for both athletes and non-athletic patients, regardless of age and sex. To achieve this goal, we must accumulate information on parameters before starting large-scale clinical trials in the near future to more accurately evaluate PRP quality and the systemic and local physical conditions of patients. In this study, at an early stage of the project, we aimed to identify the differences in the biochemical composition of leukocyte-rich PRP (L-PRP) preparations between female athletes and their non-athletic counterparts.

## 2. Results

Regarding the characteristics of the female college athletes we examined, the “parent” cohort of this study showed a low risk of low energy availability in 5 out of 71 participants, a low and medium risk of menstrual dysfunction in four and three participants, and survivors from low and medium stress fractures in 15 and two participants, respectively. However, the number of athletes showing decreased bone mineral density (BMD) as evaluated by the z-score showed that only one participant was ranked as low-risk.

To characterize the biochemical composition of L-PRP preparations in female athletes, we performed a cross-sectional cohort study comparing female sedentary controls of the same age.

The comparisons of age, weight, and body composition indices (BCIs) (body mass index (BMI), body fat percentage (BFP), skeletal muscle percentage (SMP), bone mass weight (BMW), and BMR) are shown in Figure 1. With regard to age, the athletes were significantly younger than the controls. However, we recognized both groups as belonging to the same generation as late adolescents [33]. Regarding weight, BFP, BMW, and BMR levels, the athletes had significantly higher values than the controls. The difference in BMI was not statistically significant (*p* = 0.053); however, athletes tended to have a higher BMI than the controls. In contrast, athletes had lower SMP levels than the controls.

Comparisons of blood cell counts and hematocrit (HCT) levels in whole blood samples are shown in Figure 2. White blood cell (WBC) counts were significantly higher in athletes than in controls. HCT levels were significantly lower in the athletes than in the controls. No significant differences were observed in red blood cell (RBC) and platelet (PLT) counts.

Comparisons of blood cell counts in the L-PRP preparations are shown in Figure 3. The WBC counts per milliliter and total WBC counts per L-PRP preparation were significantly higher in athletes than in controls. Instead, the RBC counts per milliliter and total RBC counts per L-PRP preparation were significantly lower in athletes than in controls. The density of individual blood cells is summarized along with the HCT values in Table 1. Because the PLTs were concentrated at similar levels, no significant differences were observed in either count.

Comparisons of the correlations between total WBC and PLT counts in the L-PRP preparations are shown in Figure 4. These correlations were positive in both groups, but the levels of correlation were moderate (R = 0.454, *p* = 0.00612) and weak (R = 0.355, *p* = 0.0546) in athletes and controls, respectively.

Comparisons of growth factors (Platelet-derived growth factor-BB (PDGF-BB), Transforming-growth factor-β1 (TGFβ1)) and cytokines (platelet factor4 (PF4)) in L-PRP preparations are shown in Figure 5. The PDGF-BB levels per PLT, TGFβ1 levels per PLT, and total TGFβ1 levels per L-PRP were significantly lower in the athletes than in the controls. No significant differences were observed in PF4 levels during conversion using PLT counts and total L-PRP.

Comparisons of cytokines related to inflammation (Interleukin-1β (IL-1β), Interleukin-1 receptor antagonist (IL-1RA)) in the L-PRP preparations are shown in Figure 6. The levels of the inflammatory cytokine IL-1β in the athletes were significantly higher than in controls in terms of conversion using PLT counts and total L-PRP. The anti-inflammatory factor IL-1RA was significantly higher in athletes than in controls at all conversions, including WBC counts.

Comparisons of the correlations between PLT counts and IL-1β or IL-1RA levels in L-PRP preparations are shown in Figure 7. All correlations were positive, and the strength ranged from weak to moderate (R = 0.260−0.474).

Comparisons of the correlations between WBC count and IL-1β or IL-1RA levels in L-PRP preparations are shown in Figure 8. As shown in Figure 7, all correlations were positive, but the strength was much higher for the WBC counts than for the PLT counts (R = 0.673–0.945). In addition, significant differences were observed in all the correlations.

## 3. Discussion

In this study, we found that, compared to controls, growth factor levels decreased and pro-inflammatory and anti-inflammatory cytokine levels increased in the L-PRP of female athletes. L-PRP was chosen for this study because it is preferably used for the treatment of anterior cruciate ligament injury, which has a higher injury rate in female athletes [34,35,36]. A previous study focused on male professional soccer players to accurately quantify PLT-associated growth factor levels [16], we used P-PRP instead of L-PRP. However, we obtained similar data and did not observe any apparent differences between male and female athletes (with respect to their corresponding controls). These phenomena can be explained by microinjuries caused by intense regular exercises and games. Although not a contact sport, muscle fibers and related cells can frequently be damaged at the micrometer level in athletes also, and tissue repair begins with the aggregation of activated PLTs. Thus, even though they float in a pebble-like shape in the circulatory system, PLTs may frequently experience reversible activation, which releases growth factors and consequently reduce storage in their short lifetime.

In contrast, cytokines related to inflammatory responses, particularly the anti-inflammatory IL-1RA, were increased in the L-PRP of athletes. Because increased anti-inflammatory factors prevent or attenuate the undesirable exaggeration of inflammation in the inflammatory phase, it could be thought that the wound healing process proceeds without delay. Thus, although this PRP preparation is inferior to other PRP preparations in terms of growth factor levels, it may not necessarily be inferior in terms of tissue regeneration potency. Further investigation is needed to reveal the effects of this difference on clinical outcomes. However, we suggest that pro-inflammatory and anti-inflammatory cytokines should be quantified as significant parameters before performing clinical trials, as previously proposed [37].

### 3.1. Historical Background of PRP Therapy and Issues Remaining to Be Addressed

PRP therapy is used during the healing process of the body [2]. Injection or implantation of PRP at the injury site is expected to reproduce or facilitate this scenario, mainly through the growth factors stored in PLTs. In ideal cases, PRP is thought to positively affect inflammation, angiogenic processes, anabolism-catabolic balance, and alter the existing microenvironment at the injury site [38]. In the 1990s, skeletal regeneration after PRP treatment was first reported [39]. Due to its high safety and cost-effectiveness ratio, PRP therapy has spread worldwide, particularly in countries with higher medical care expenditures. However, to the best of our knowledge, PRP has rarely been studied in basic or preclinical studies as a newly developed drug [40].

Nonetheless, clinical studies and clinical practices precede without paying much attention to unsuccessful cases [41]. Consequently, many clinical questions and skepticism have emerged. Thus, the current controversial situation could originate from such an exceptional history of R&D in its early stages. To make the upcoming large-scale RCTs more reliable and fruitful, PRP-related factors such as variations in preparations and dosage; patient-related factors such as age, sex, disease history, and treatment history of patients [42]; and treatment protocols should be standardized or classified to easily evaluate the correlation with clinical outcomes. In this study, we used only L-PRP, based on clinical preferences. However, the inclusion or exclusion of leukocytes is controversial issue in this field [7,43].

### 3.2. Possible Specificity of the Young Female Athlete Besides PRP Compositions

In the field of sports science and medicine, male athletes have always been dominant and have therefore been investigated vigorously. Although the number of female athletes participating in sports continues to increase, data on injuries in female athletes are often underrepresented.

The FAT has been indicated as the major difference in physical condition between male and female athletes [22,24,44,45]. It is defined as a combination of disordered eating, amenorrhea, and osteoporosis, and is considered a key symptom in maintaining the health of young female athletes. Most female college athletes appeared to be healthy in terms of energy availability, menstrual function, and bone health. These parameters, particularly those related to “low energy availability” in FAT and skeletal muscle mass in BCIs, indicate the ability to deliver nutrients, oxygen, and circulating stem cells to peripheral tissues and, therefore, should be considered key patient-related parameters influencing the clinical outcome of PRP treatment.

### 3.3. Limitations

However, this pilot study requires further investigations. Firstly, the sample size was too small to reach firm conclusions. Although these tendencies are reproducible, the absolute values of the individual parameters cannot be considered reproducible. Secondly, blood collection from the same donor was not repeated to determine the reproducibility. Regarding the levels of growth factors associated with platelets, pure PRP preparations are better suited for obtaining reproducible data. Because L-PRP is frequently used in the treatment of injured female athletes in affiliated hospitals, we chose L-PRP for subsequent analyses. Thirdly, the BCI data were neither accurate nor absolute. Assuming that the BCIs were measured upon blood collection in related hospitals, we adopted a bioelectrical impedance analysis (BIA) technique to conveniently measure body composition indices. However, BIA is inferior to densitometry and other X-ray-based techniques in terms of its accuracy and reliability. Thus, these data should be compared among groups measured using the same instrument.

## 4. Materials and Methods

### 4.1. Participants and Study Design

A cross-sectional study was performed in two independent groups of healthy female participants: one (non-athlete sedentary: control) was composed of ordinary healthy adults (age: 20–25), while the other (athlete) was composed of college athletes (age: 18–22) who received dietary counseling. The inclusion criteria for the control group were as follows: healthy late-adolescent females who were nonsmokers had no systemic diseases regardless of medical control, underwent no daily physical training, and agreed to provide informed consent. Exclusion criteria included acute or chronic inflammatory conditions reflected in blood cell counts and current or former thrombotic or PLT disorders. The inclusion and exclusion criteria for the athlete group were identical to those for the control group, with an additional criterion of continuous daily physical training.

### 4.2. Blood Collection and Preparation of L-PRP

Blood was collected from participants between meals, i.e., not within approximately 1 h of breakfast or lunch, in glass vacuum tubes containing acid-citrate-dextrose (ACD-A) (Vacutainer, Becton, Dickinson, and Company, Franklin Lakes, NJ, USA), as described previously [46]. Whole blood samples were transported from the hospitals to the laboratory by a parcel delivery service at ambient temperatures (approximately 3–10 °C). Before preparing L-PRP, the samples were prewarmed for 2 h at 20–25 °C to restore the PLTs to the resting state. L-PRPs were prepared using the double-spin method. Briefly, blood samples were centrifuged at 738× *g* for 10 min, and the buffy coat was collected at the bottom of the platelet-poor plasma fraction and the upper RBC fraction (Figure 9). The second centrifugation was performed at 664× *g* for 3 min to adjust the volume of L-PRP to approximately 0.9 mL. Both preparations were not activated and stored at −80 °C until use [46].

The workflow diagram from weighing to L-PRP preparation is depicted in Figure 9.

### 4.3. Blood Cell Counting

Blood cell counts were performed using an automated hematology analyzer (pocH iV-diff, Sysmex Corporation, Kobe, Japan) [16]. In addition to cell counting, data on HCT was obtained.

Cell counting in whole blood samples was expressed as concentration (cell counts per μL). In L-PRP preparations, the cell counting data was expressed as both concentration and total cell counts in L-PRP preparations (per L-PRP) because total number of cells, which is equal to “dose” in medication, is more important to evaluate clinical outcomes.

### 4.4. Determination of Growth Factor and Cytokine Levels by ELISA

As previously described [16], the concentrations of PDGF-BB, platelet factor 4 (PF4), TGF-β1, IL-1β, and IL-1RA in frozen L-PRP were determined using human PDGF-BB, TGF-β1, PF4, IL-1β, and IL-1RA Quantikine ELISA kits (R&D Systems, Inc., Minneapolis, MN, USA).

As in the data expression in blood cell counting, the data for growth factors or cytokines were normalized by both PLT or WBC count (per PLT) or expressed as total amounts in L-PRP preparations (per L-PRP). Since these bioactive factors are derived from PLT or WBC, it is important to compare the amounts per unit cell number. However, for the reasons described in the above subsection, it is also important to record the total amounts of bioactive factors to evaluate clinical outcomes. For reference, we express the switching of the data expression manner as “conversion”.

### 4.5. Determination of Body Composition

Before blood collection, body composition of the participants was determined using a bathroom weighing scale (HCS-FS03; ECLEAR, ELECOM). This scale was installed with a unique MRI-based program that enables a more accurate evaluation of individual body fat percentages (BFP) based on measured bioelectrical impedance and body weight [47]. BMI, BFP, SMP, BMW, and BMR were automatically determined using this weight scale.

### 4.6. Questionary Survey for Female Athlete Triad

A survey of the presence of the female athlete triad was conducted using a short questionnaire (Table 2).

### 4.7. Statistical Analysis

The data is expressed as means ± SD in Table 1.

To compare each index between the two groups, data were expressed as box plots, and the Mann–Whitney U test, Welch’s *t*-test, or Student’s *t*-test were performed to confirm statistical differences in the median and spread (SigmaPlot version 14.5; Systat Software, Inc., Palo Alto, CA, USA). In box and whisker plots, which are usually designated as box plots, we express the distribution of data based on a five-number summary: (1) minimum value, (2) first quartile, (3) median, (4) third quartile, and (5) maximum value. In addition, we display the outliers using filled circles.

Pearson’s correlation analysis was performed to compare the correlation between the two indices, and the correlation coefficients were calculated using SigmaPlot software. Differences were considered statistically significant at *p*  <  0.05. The strength of the correlation was defined as very strong (0.8–1.0), strong (0.6–0.79), moderate (0.4–0.59), weak (0.2–0.39), and very weak (0–0.19).

## 5. Conclusions

The biochemical compositions of female college athletes who continued tough regular exercise and received dietary counseling were compared with those of female sedentary controls of the same generation. PDGF-BB and TGFβ1 levels were significantly lower in athletes than in controls, while IL-1β and IL-1RA levels were significantly higher in athletes than in controls. These findings suggest that L-PRP in athletes may control the initial inflammatory phase of the tissue regeneration process, possibly contributing to favorable clinical outcomes.

This conclusion does not directly support the possibility of higher effectiveness of PRP treatment in female athletes. However, to test this possibility in future clinical trials, we recommend determining anti-inflammatory factor levels as significant parameters along with physical parameters related to BCIs and FAT before starting large-scale randomized clinical trials.

## Figures and Tables

**Figure 1 ijms-24-13592-f001:**
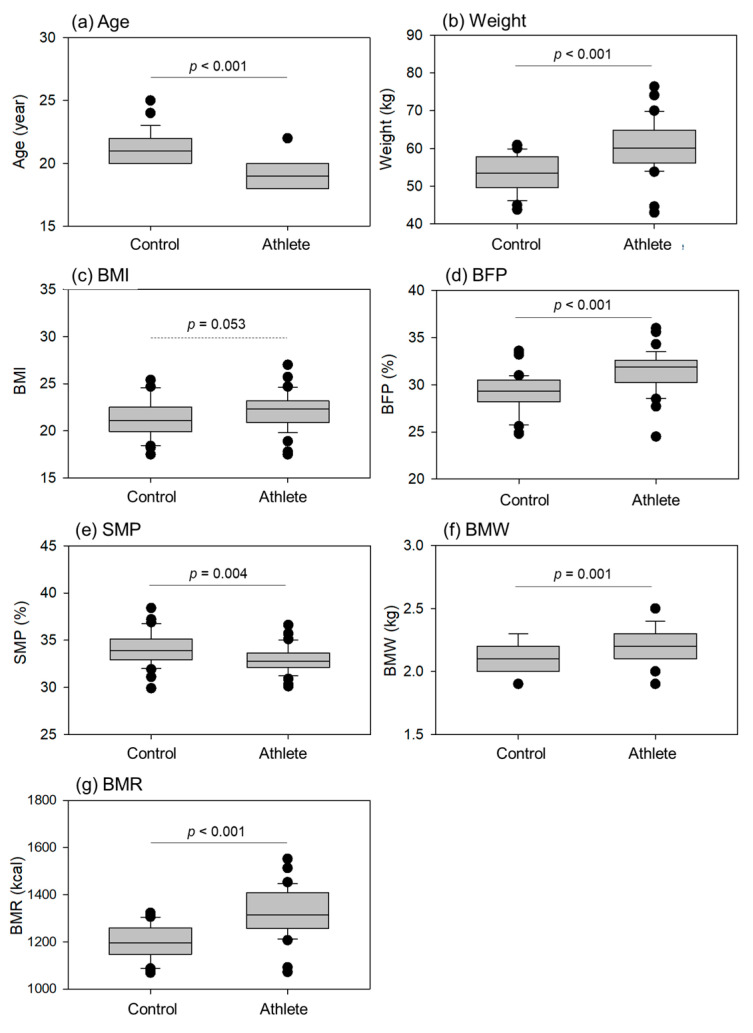
Age, weight, and body composition indices (BCIs), such as body mass index (BMI), body fat percentage (BFP), skeletal muscle percentage (SMP), bone mass weight (BMW), and basal metabolic rate (BMR) in the control (n = 30) and athlete groups (n = 35). Statistical analyses were performed using the Mann–Whitney U test (**a**,**d**,**f**) or Student’s *t*-test (**b**,**c**,**e**,**g**). Statistical differences in age (**a**), weight (**b**), BFP (**d**), SMP (**e**), BMW (**f**), and BMR (**g**) were observed as each *p* value is shown in each panel.

**Figure 2 ijms-24-13592-f002:**
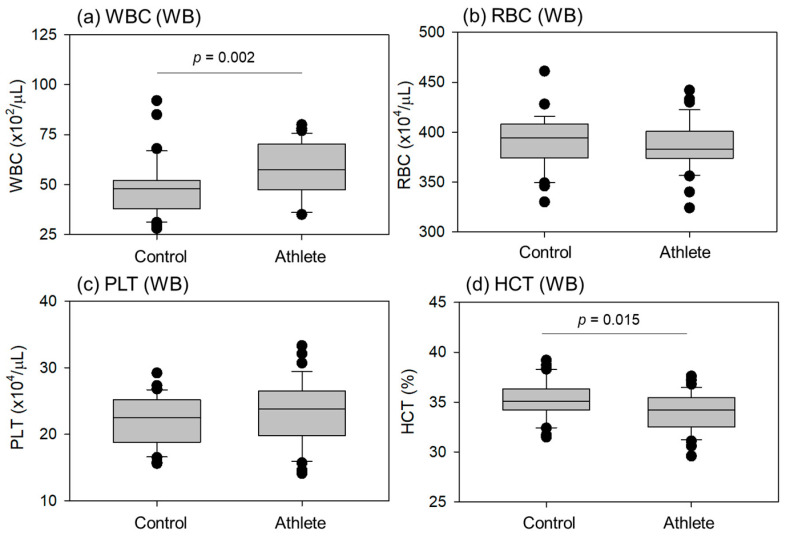
Blood cell counts and hematocrit (HCT) levels in whole-blood (WB) samples from the control (n = 30) and athlete (n = 35) groups. Statistical analyses were performed using the Mann–Whitney U test (**a**) or Student’s *t*-test (**b**–**d**). Statistically significant differences were observed between the WBC (**a**) and HCT (**d**) as each *p* value is shown in each panel.

**Figure 3 ijms-24-13592-f003:**
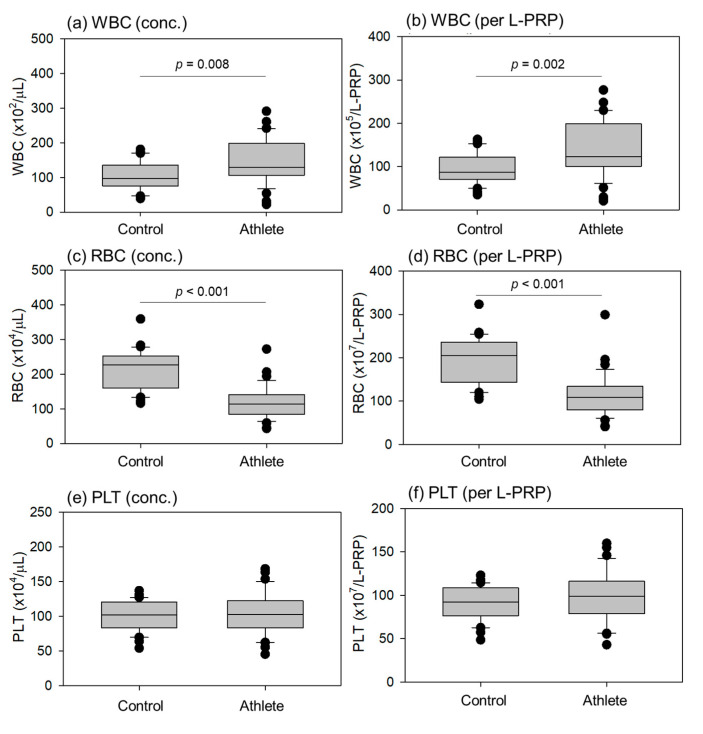
Blood cell counts in leukocyte-rich PRP (L-PRP) preparations of the control (n = 30) and athlete (n = 35) groups. The concentrations of white blood cell (WBC), red blood cell (RBC), and platelet (PLT) are depicted in panels (**a**,**c**,**e**), respectively, while the total cell counts per L-PRP preparation are shown in panels (**b**,**d**,**f**). Statistical differences were observed in WBC (**a**,**b**) and RBC (**c**,**d**) counts as each *p* value is shown in each panel.

**Figure 4 ijms-24-13592-f004:**
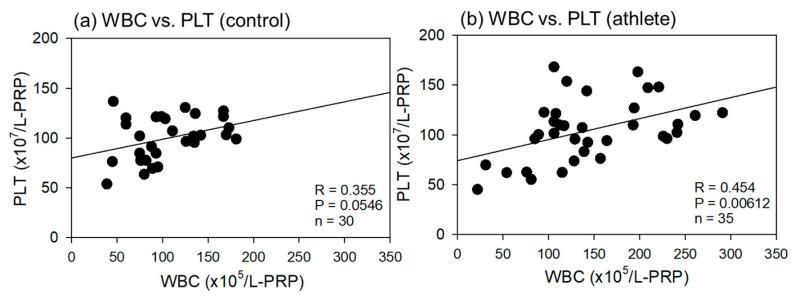
Correlations between total white blood cell (WBC) and platelet (PLT) counts in leukocyte-rich PRP (L-PRP) preparations of the control (**a**) (n = 30) and athlete groups (**b**) (n = 35). The strength of the associations was analyzed using Pearson’s correlation coefficient. The strength of the correlation was defined as very strong (0.8–1.0), strong (0.6–0.79), moderate (0.4–0.59), weak (0.2–0.39), and very weak (0–0.19).

**Figure 5 ijms-24-13592-f005:**
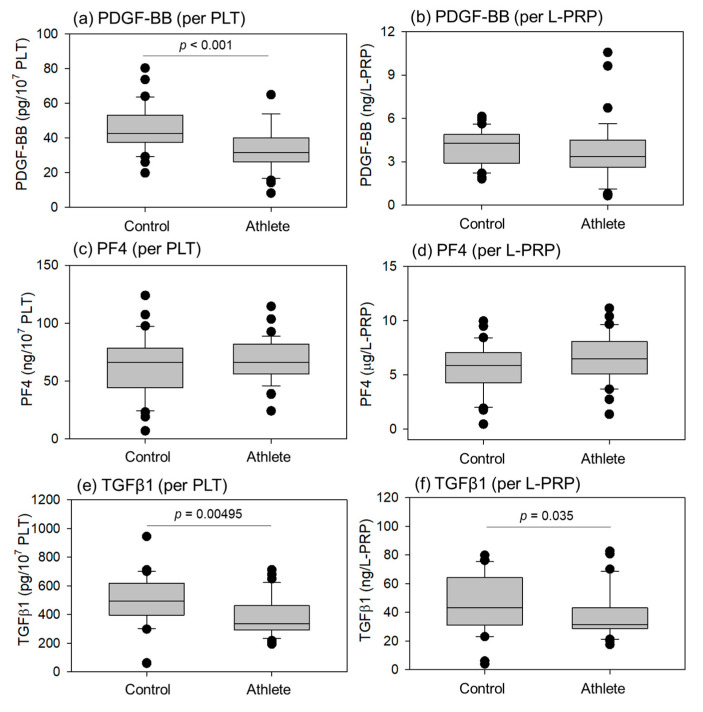
Levels of PDGF-BB (**a**,**b**), PF4 (**c**,**d**), and TGFβ1 (**e**,**f**) in L-PRP preparations of the control (n = 30) and athlete groups (n = 35). Statistical analyses were performed using the Mann–Whitney U test (**a**,**b**,**f**), Welch’s *t*-test (**c**), or Student’s *t*-test (**d**,**e**). Each *p* value is shown in each panel.

**Figure 6 ijms-24-13592-f006:**
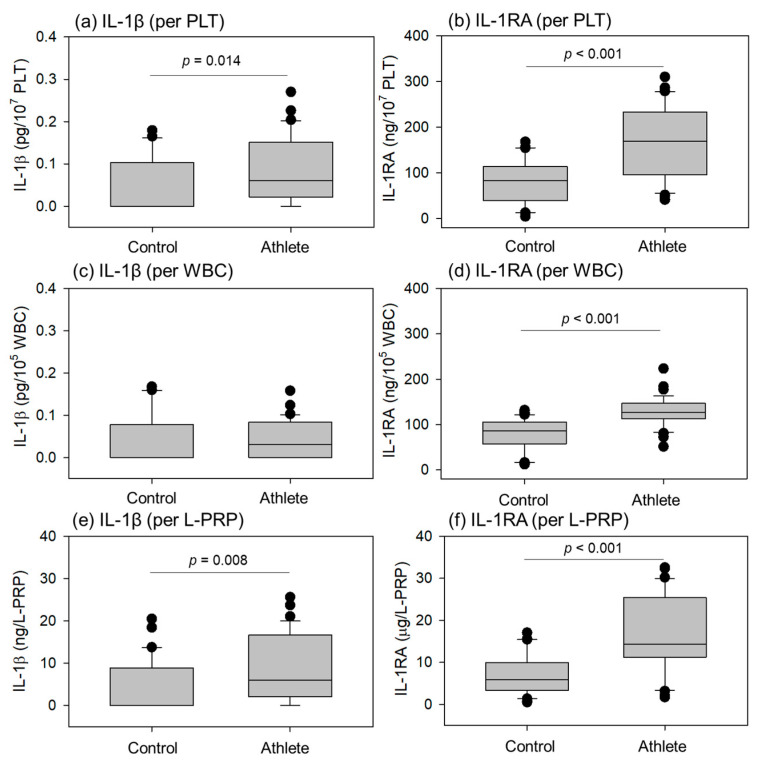
Levels of interleukin-1β (IL-1β) (**a**,**c**,**e**) and IL-1 receptor antagonist (IL-1RA) (**b**,**d**,**f**) in leukocyte-rich PRP (L-PRP) preparations of the control (n = 30) and athlete groups (n = 35). Each level was normalized by platelet (PLT) count (**a**,**b**), white blood cell (WBC) count (**c**,**d**), or L-PRP preparation (**e**,**f**). Statistical analyses were performed using the Mann–Whitney U test (**a**,**e**,**f**), Welch’s *t*-test (**b**,**d**), or Student’s *t*-test (**c**). Each *p* value is shown in each panel.

**Figure 7 ijms-24-13592-f007:**
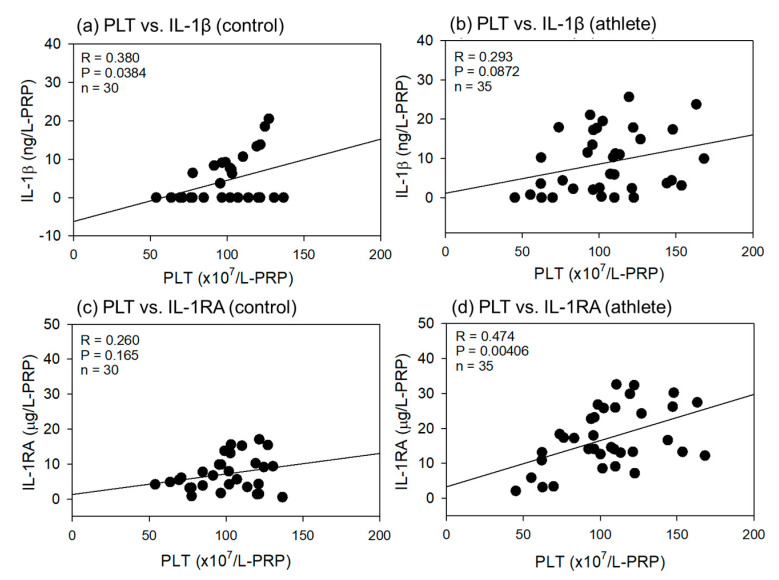
Correlations between platelet (PLT) counts and interleukin-1β (IL-1β) (**a**,**b**) or IL-1 receptor antagonist (IL-1RA) (**b**,**d**) in leukocyte-rich PRP (L-PRP) preparations of the control (**a**,**c**) (n = 30) and athlete groups (**b**,**d**) (n = 35). The strength of the associations was analyzed by the Pearson correlation coefficient. The strength of the correlation was defined as very strong (0.8–1.0), strong (0.6–0.79), moderate (0.4–0.59), weak (0.2–0.39), and very weak (0–0.19).

**Figure 8 ijms-24-13592-f008:**
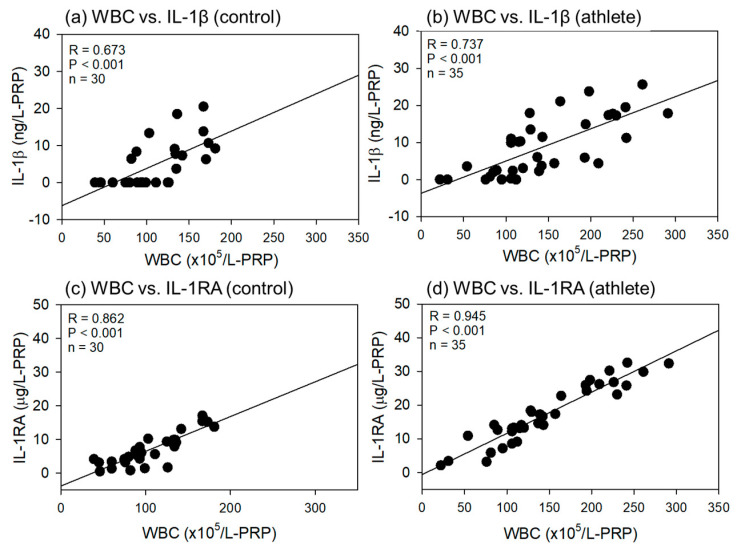
Correlations between white blood cell (WBC) counts and interleukin-1β (IL-1β) (**a**,**b**) or IL-1 receptor antagonist (IL-1RA) (**b**,**d**) in L-PRP preparations of the control (**a**,**c**) (n = 30) and athlete groups (**b**,**d**) (n = 35). The strength of the associations was analyzed by the Pearson correlation coefficient. The strength of the correlation was defined as very strong (0.8–1.0), strong (0.6–0.79), moderate (0.4–0.59), weak (0.2–0.39), and very weak (0–0.19).

**Figure 9 ijms-24-13592-f009:**
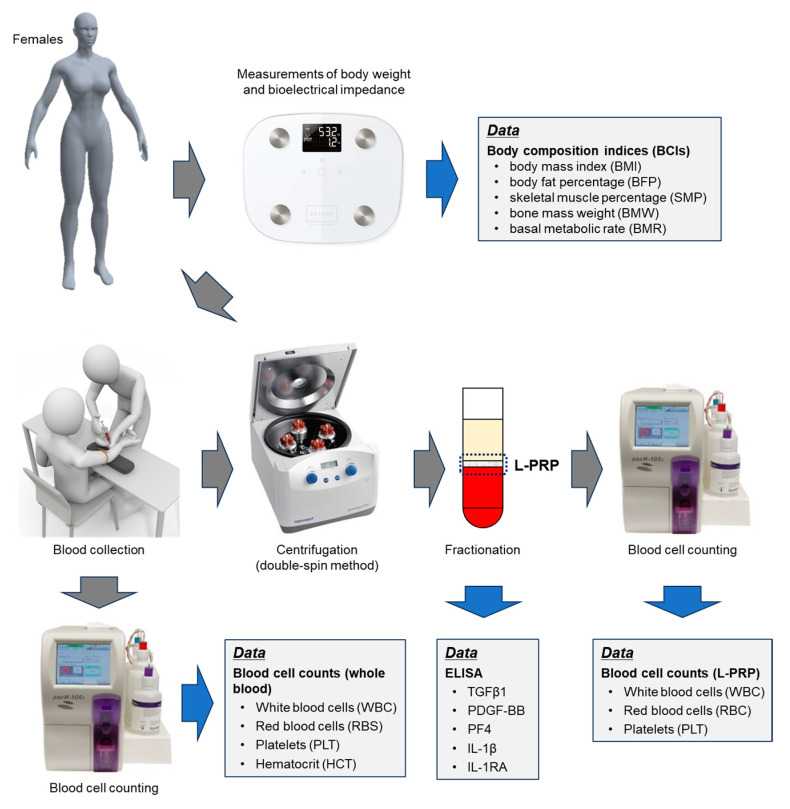
The workflow diagram of this study.

**Table 1 ijms-24-13592-t001:** Blood cell counts and hematocrit before and after L-PRP preparation.

Indices	Control			Athlete		
	WB	L-PRP	Folds	WB	L-PRP	Folds
WBC	48.0 ± 14.7 (×10^2^/μL)	106.6 ± 40.8 (×10^2^/μL)	2.22	57.5 ± 13.2 (×10^2^/μL)	144.8 ± 66.3 (×10^2^/μL)	2.52
RBC	391.5 ± 26.4 (×10^4^/μL)	210.4 ± 58.2 (×10^4^/μL)	0.537	385.9 ± 24.4 (×10^4^/μL)	118.2 ± 48.4 (×10^4^/μL)	0.306
PLT	22.0 ± 3.7 (×10^4^/μL)	100.1 ± 21.6 (×10^4^/μL)	4.55	23.4 ± 4.8 (×10^4^/μL)	104.6 ± 30.8 (×10^4^/μL)	4.47
HCT	35.1 ± 2.0 (%)	N.D.	N.D.	33.9 ± 2.0 (%)	N.D.	N.D.

The data is expressed as the mean ± SD of 30 (control) or 35 samples (athlete). WB: whole blood, L-PRP: leukocyte-rich PRP, WBC: white blood cell, RBC: red blood cell, PLT: platelet, HCT: hematocrit, N.D.: not determined.

**Table 2 ijms-24-13592-t002:** Summary of the questionnaire.

Triad	Questionnaire
Amenorrhoea	Have you experienced no menstruation for more than three months?
	Have you experienced fewer than five menstrual cycles in the previous year?
Bone strength	Have you ever been diagnosed with osteoporosis?
Low energy availability	Is your BMI 17.5 or less?

## Data Availability

Data are available from the corresponding author upon request.

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
