# Peer review of "Characterization of Leukocyte- and Platelet-Rich Plasma Derived from Female Collage Athletes: A Cross-Sectional Cohort Study Focusing on Growth Factor, Inflammatory Cytokines, and Anti-Inflammatory Cytokine Levels"

_ijms, 2023, doi:10.3390/ijms241713592_

Round 1
Reviewer 1 Report
It is very important study which may create new avenues for therapy.
Author Response
It is very important study which may create new avenues for therapy.
Response: Thank you for your evaluation.
Reviewer 2 Report
The manuscript have a good result but it does not present in a clear way. The author should present the figures in a more clear and thoughtful way so the reader can understand without going back and forth of the texts in the manuscript. Add necessary Abbreviations in Figures, add Statistical significance information as well. And The strength of the correlation was defined as very strong (0.8−1.0), strong (0.6−0.79), 287 moderate (0.4−0.59), weak (0.2−0.39), and very weak (0−0.19) to figure legends.
What is the function of the questionnaire? If necessary, L 277, should the questionnaire show as a table or supplementary information?
One schematic of the workflow would help reader understand better the scheme of the manuscript.
The author need to point that which of the value of P Statistical significance, were considered significant? p < 0.05 (*) or p < 0.01 (**) ?
How can this study avoid the risk of bias? And have the author also considering the habit factor? As I know, exposure to alcoholic beverages, eating habits could also affect the level of inflammation factors.
In addition, the author could add more discussion on the contribution of this study and may generate a new hypothesis for the future large clinical study.
There are also some format errors so the author should read through the manuscript again and revise it.
L202, delete Platelet-rich plasma
PRP as newly developed drug?
L224, data are should be data is
L226, female athlete triad should be FAT
Add space between number and unit, such as 10°C should be 10 °C, n=35 should be n = 35
In abstract, the abbreviation and the full description should display at the first place, such as Interleukin-1β should be Interleukin-1β (IL-1β), Transforming-growth factors β1 etc.
More reference in L226
I am confused with the text in L82-85
Moderate editing of English language required.
There are some texts which need to rewrite or edit, such as
L35, edit
L194-197, rewrite
L298-300, rewrite
L200, give more information or explanation
Author Response
- The manuscript have a good result but it does not present in a clear way. The author should present the figures in a more clear and thoughtful way so the reader can understand without going back and forth of the texts in the manuscript. Add necessary Abbreviations in Figures, add Statistical significance information as well. And The strength of the correlation was defined as very strong (0.8−1.0), strong (0.6−0.79), 287 moderate (0.4−0.59), weak (0.2−0.39), and very weak (0−0.19) to figure legends.
Response: Based on your advice, we modified the data presentation. We hope that this revision makes the figures clearer and more thoughtful.
Regarding abbreviations, statistical significance information and the strength of the correlation were added to the figure legends.
- What is the function of the questionnaire? If necessary, L 277, should the questionnaire show as a table or supplementary information?
Response: Thank you for this comment. We aimed to determine whether this cohort was suitable for the analysis of PRP quality and the relationship between PRP quality and body composition indices. If this cohort shows a typical “female athlete triad,” we thought that we had to consider many more factors influencing PRP quality and choose appropriate counterparts. However, as described in the Discussion section (#3.2), we did not find appreciable differences between the athlete and non-athlete groups. To help readers understand, we added a table summarizing the questionnaire (Table 2).
- One schematic of the workflow would help reader understand better the scheme of the manuscript.
Response: Thank you for this advice. We added the workflow diagram in Figure 9.
- The author need to point that which of the value of P Statistical significance, were considered significant? p < 0.05 (*) or p < 0.01 (**) ?
Response: Thank you for this comment. As described in the Materials and Methods section (#4.7), we considered “P < 0.05” statistically significant. To show the probability levels in more detail, however, we put the labels of P < 0.01 and others in graphs.
- How can this study avoid the risk of bias? And have the author also considering the habit factor? As I know, exposure to alcoholic beverages, eating habits could also affect the level of inflammation factors.
Response: Thank you for this advice. We agree with the reviewer’s suggestion. However, as described in the Materials and Methods section (#4.1), female college athletes received dietary counseling. Patients with symptoms and medical history related to acute and chronic inflammation were excluded. Thus, even though such a risk of bias was not completely eliminated, we believe that it was minimized.
- In addition, the author could add more discussion on the contribution of this study and may generate a new hypothesis for the future large clinical study.
Response: Thank you for this comment. As described in the Introduction section, there are few publications regarding the comparison between athletes and non-athletes in females, although comparisons between male and female athletes have often been reported. Therefore, it is necessary to first accumulate and analyze data from female athletes (vs. their female counterparts).
We believe that this effort will eventually contribute to the evaluation of the clinical outcomes of PRP therapy and the improvement of PRP therapy to prolong athlete careers.
In this revised version, according to your advice, we have included a table, figure, and some modifications in the text, which expanded the length of the manuscript considerably. Thus, in this review, we decided not to further expand the Discussion section.
- There are also some format errors so the author should read through the manuscript again and revise it.
Response: Thank you for this indication. Because we used a template, we thought that the format was correct. We did not find any format errors in the manuscript downloaded from the submission site. We hope that this has been corrected by editorial staff.
- L202, delete Platelet-rich plasma. PRP as newly developed drug?
Response: Thank you for this comment. We deleted the full name “Platelet-rich plasma” but left “PRP” there. This expression may have caused misunderstanding. As explained in detail in our previous review article [Kawase and Okuda, Biomed Res Int, 6389157; 2018], PRP was not investigated in the established way that is applied in the process of approval of “new drugs.” New drugs are usually tested in preclinical studies (from in vitro to animal experiments) and then subjected to further testing in clinical trials (from phases I to IV). However, PRP has been used directly in clinical settings without any preclinical or clinical studies. We emphasize that this irregular R&D process causes confusion. We have included this reference.
- L224, data are should be data is
Response: Thank you for this comment. Not only in this sentence but also in other sentences including “data,” we replaced “are” and “were” with “is” and “was.”
- L226, female athlete triad should be FAT
Response: Thank you for this comment. We added the abbreviation “FAT” here.
- Add space between number and unit, such as 10°C should be 10 °C, n=35 should be n = 35
Response: Thank you for this comment. We modified these expressions.
- In abstract, the abbreviation and the full description should display at the first place, such as Interleukin-1β should be Interleukin-1β (IL-1β), Transforming-growth factors β1 etc.
Response: Thank you for this comment. We described both the full descriptions of the cytokines and their abbreviations in the abstract.
- More reference in L226
Response: Thank you for this comment. We added three more references here.
- I am confused with the text in L82-8
Response: Thank you for this comment. As explained in the response to your early comment, there are a sufficient number of publications reporting comparisons between male and female athletes but rarely between female athletes and their counterparts. Although we may not have mentioned what we wanted to explain there, we hope that you will understand the aim of this paragraph. It is difficult to expand this paragraph without a sufficient number of studies.
- Comments on the Quality of English Language
- Moderate editing of English language required.
- There are some texts which need to rewrite or edit, such as
Response: Thank you for this comment. We have ordered professional Editage English editing service to meet your request.
- L35, edit
- L194-197, rewrite
- L298-300, rewrite
- L200, give more information or explanation
Response: Thank you for these comments.
L35 (L35-37), we modified this sentence.
L194-197 (L216-221), we rewrote this sentence.
L298-300 (L347-350), we extensively rewrote this sentence.
L200 (L223-224), we cited our recent publications that mentioned in more detail what we aim for in future clinical trials.
Reviewer 3 Report
Tomoharu Mochizuki et al.'s article covers a topic that falls within the scope of the journal. The article is both timely and pertinent. Several aspects should be improved by the authors in order to publish it. One of the weakest points is the discussion. I am suggesting a few points to improve the paper:
· In references, please abbreviate the Journal name of ref #5 (De Macedo, A.; Lana, J.; Pedrozo, C.; Bottene, I.; De Medeiros, J.; Da Silva, L. The Regenerative Medicine Potential of PRP in Elite Athlete Injuries. Fortune Journal of Rheumatology. 2020, 2, 16-26).
· In reference #12, the “page number” is missing (article number 1835)
· Please, in line 70, include de meaning of “BMR”, as it is the first time mentioned.
· It would be useful to include an abbreviation list in the paper.
· Line 252. Please, specify the University of…
· Line 255. “Blood was collected from participants between meals”. Fasting?
· Line 261. PBS: Phosphate-buffered saline?
· Line 373. “’” Typo.
· Point. 4.2 Blood collection and preparation of L-PRP. Please, expand the protocol of L-PRP obtention (it is referenced as #24, but it would be more useful and friendly to explain it in the paper) (centrifugation time and speed,…). Please, also confirm if the samples were or no activated)
· Point 4.5 Determination of body composition. Please specify how the body composition was determined. With a bathroom weighing scale OR with MRI? In case of MRI measures, please specify the image acquisition and treatment parameters.
· Point 4.6 Questionary survey for female athlete triad. Please reference or indicate the questionnaire.
· Line 283: Systat Software is repeated. Also specify the country.
· For ease of reading, please specify abbreviations in the figure captions or in the figures themselves.
· “the athletes were significantly younger than the controls”. Age influence in the results? Please discuss. Some GGFF depends on the age ( for example IGF…).
· To enhance the value of the paper, it would be worthwhile to include a table showing PLA, WBC, ERI, HTC values for both peripheral blood and L-PRP. Additionally, include the increase in platelets and leukocytes values compared to peripheral blood (either in fold or increase).
· One question to be resolved is why athletes have a higher percentage of body fat and less muscle than sedentary people. Please confirm these findings, and if they are correct, explain what you think the reasons might be, as it is counter-intuitive.
· In the figures, specify whether the box-and-bigot plot indicates the median or the mean, and the confidence interval at ...%. Are the dot outliers. Please specify.
· How do the authors explain that the hematocrit is higher in sedentary women than in female athletes?
· Figure 3. It is not clear to this reviewer how the values in the right column panels (b, d, and f) are calculated. Please specify in M&M. I understand that the values in a, c, and e are concentration values, and those in b, d, and f are total values, multiplying the concentration by the volume of L-PRP? Please specify for the sake of clarity (also fig. 5 and others). For example, in the figure 6 it is partially explained “Each level was normalized by PLT count (a, b), WBC count (c, d), or L-PRP preparation (e, f)” but sorry, I don’t understand the normalization of L-PRP (e & f).
· Lines 145-146: “No significant differences were observed in PF4 levels between the conversions.” Clarify this sentence. What are the “conversions”? Also in lines 154 and 156.
· Point 3. Discussion. The second sentence (lines 180-182) states that L-PRP is more effective for female athletes in preventing ACL injuries than P-PRP. Although the authors refer to a paper that provides the incidence date, they do not present any data confirming that L-PRP is more effective than P-PRP. Could you please provide further clarification?
· Line 186. “These phenomena…” Kindly explain the phenomena to which the authors refer.
· Lines 209-210. Please clarify this sentence: “However, to our perplexity, PRP has rarely been studied either in basic or preclinical studies as a newly developed drug.”
· Lines 229-234. The data provided here should be in the results, and not be used for the first time in the discussion.
· Overall, the discussion is poor, as it does not really go into depth on the results obtained in comparison with other works. No explanations are provided on the differences observed between the two study groups. The authors have several questions that I ask in this review, to deepen the discussion, but there are many more.
· In the discussion, it is important to include a paragraph explaining the limitations of the study.
· Conclusions: Lines 294-296. “These findings suggest that although L-PRP may be less potent in accelerating cell growth, it may be able to control the initial inflammation phase to facilitate tissue regeneration.” Less potent than…? Less potent in athletes than controls? Please, only make conclusions supported by the obtained results.
Moderate editing of English language required
Author Response
- Tomoharu Mochizuki et al.'s article covers a topic that falls within the scope of the journal. The article is both timely and pertinent. Several aspects should be improved by the authors in order to publish it. One of the weakest points is the discussion. I am suggesting a few points to improve the paper:
- In references, please abbreviate the Journal name of ref #5 (De Macedo, A.; Lana, J.; Pedrozo, C.; Bottene, I.; De Medeiros, J.; Da Silva, L. The Regenerative Medicine Potential of PRP in Elite Athlete Injuries. Fortune Journal of Rheumatology. 2020, 2, 16-26).
Response: Thank you for this advice. We replaced the full name of this journal with its abbreviation.
- In reference #12, the “page number” is missing (article number 1835)
Response: Thank you for this comment. We added the page number of this reference.
- Please, in line 70, include de meaning of “BMR”, as it is the first time mentioned.
Response: Thank you for this advice. We corrected it.
- It would be useful to include an abbreviation list in the paper.
Response: Thank you for this advice. We added the list of abbreviations.
- Line 252. Please, specify the University of…
Response: Thank you for this comment. We added the full name of the university.
- Line 255. “Blood was collected from participants between meals”. Fasting?
Response: Thank you for this comment. This did not indicate fasting. We avoided blood collection within approximately 1 h of breakfast or lunch. We added this annotation to the revised manuscript.
- Line 261. PBS: Phosphate-buffered saline?
Response: Thank you for this comment. We added the full name of PBS in the text and inserted PBS into the abbreviation list.
- Line 373. “’” Typo.
Response: Thank you for this advice. We corrected this typo.
- 4.2 Blood collection and preparation of L-PRP. Please, expand the protocol of L-PRP obtention (it is referenced as #24, but it would be more useful and friendly to explain it in the paper) (centrifugation time and speed,…). Please, also confirm if the samples were or no activated)
Response: Thank you for this advice. We added a brief description of the procedure there.
- Point 4.5 Determination of body composition. Please specify how the body composition was determined. With a bathroom weighing scale OR with MRI? In case of MRI measures, please specify the image acquisition and treatment parameters.
Response: Thank you for this question. As described above, the advantage of this bathroom weighing scale is that a unique algorithm was developed based on the data obtained from a large population of Japanese people who underwent MRI examinations. Thus, although MRI examinations were not performed on individual participants in this study, this MRI-based program is expected to enable a more accurate evaluation of individual body fat percentage (BFP) based on the measured bioelectrical impedance and body weight.
- Point 4.6 Questionary survey for female athlete triad. Please reference or indicate the questionnaire.
Response: Thank you for this comment. We added the summary of the questionnaire in Table 2.
- Line 283: Systat Software is repeated. Also specify the country.
Response: Thank you for this advice. We corrected it and added the location of the headquarter.
- For ease of reading, please specify abbreviations in the figure captions or in the figures themselves.
Response: Thank you for this advice. We added the full names of the abbreviations used in the figures in the figure captions.
- “the athletes were significantly younger than the controls”. Age influence in the results? Please discuss. Some GGFF depends on the age ( for example IGF…).
Response: Thank you for this comment. The effects of age on the growth factor content have often been reported. Thus, we recognize its influence on the data. However, in our understanding, these studies usually compared different generations, for example, those in their 40s and 60s. In this study, we obtained the statistical difference between the athlete and control groups; however, the averages were 21.2 ± 1.3 vs. 19.1 ± 0.9. We found that both groups were included in the same generation even though they were in late adolescence, based on the assumption that sports experience is more influential than the age of this difference level. Thus, to avoid confusing readers and focus on the main theme, we decided not to expand the discussion about the age of this time.
- To enhance the value of the paper, it would be worthwhile to include a table showing PLA, WBC, ERI, HTC values for both peripheral blood and L-PRP. Additionally, include the increase in platelets and leukocytes values compared to peripheral blood (either in fold or increase).
Response: Thank you for this advice. We added this data in Table 1.
- One question to be resolved is why athletes have a higher percentage of body fat and less muscle than sedentary people. Please confirm these findings, and if they are correct, explain what you think the reasons might be, as it is counter-intuitive.
Response: Thank you for this comment. As you felt, when we first found this data, we were surprised and doubted its reliability. We were convinced that female athletes received dietary counseling to prevent eating disorders and bone fractures. In contrast, their sedentary counterparts did not have the chance to specifically learn about nutrition and sports science. In general, their counterparts did not eat as much as the athletes did. Thus, although the skeletal muscle mass was increased by dairy exercise, body fat mass was also increased by their “diet remedy.” As a result, we speculated that body fat percentage could be higher in athletes than in their counterparts.
- In the figures, specify whether the box-and-bigot plot indicates the median or the mean, and the confidence interval at ...%. Are the dot outliers. Please specify.
Response: Thank you for this advice. We have added the composition of the box plot presented in this study in the Materials and Methods section (#4.7).
- How do the authors explain that the hematocrit is higher in sedentary women than in female athletes?
Response: Thank you for this comment. We did not have any evidence or information to discuss this result. In our clinical experience, female athletes usually show a relatively lower hematocrit than non-athletic females in this generation. However, we did not attempt to clarify the possible reasons underlying this difference because the athletes were not diagnosed with anemia.
- Figure 3. It is not clear to this reviewer how the values in the right column panels (b, d, and f) are calculated. Please specify in M&M. I understand that the values in a, c, and e are concentration values, and those in b, d, and f are total values, multiplying the concentration by the volume of L-PRP? Please specify for the sake of clarity (also fig. 5 and others). For example, in the figure 6 it is partially explained “Each level was normalized by PLT count (a, b), WBC count (c, d), or L-PRP preparation (e, f)” but sorry, I don’t understand the normalization of L-PRP (e & f).
Response: Thank you for this question. This is related to your question about “conversion.” In response to your comment below, we have explained that blood cell counts and growth factor levels should be quantified not only by the concentration but also by the total amount injected into the injury site. This is because the concentration can characterize PRP preparation but is not directly linked to the clinical outcomes. The total number of platelets and growth factors is expected to directly influence clinical outcomes. Thus, we believe that these data should be expressed in two ways. The graphs in the right column of Figure 3 represent the total counts of blood cells in the individual L-PRP preparations.
- Lines 145-146: “No significant differences were observed in PF4 levels between the conversions.” Clarify this sentence. What are the “conversions”? Also in lines 154 and 156.
Response: Thank you for this comment. We adopted two units for data normalization because the total amounts (or counts) are sometimes more important than the concentrations in clinical settings. Thus, the “conversion” represents the manner of data normalization, and this term is left in the text. Instead, the explanation of the term “the conversions using PLT counts and total L-PRP” was added to its first appearance.
- Point 3. Discussion. The second sentence (lines 180-182) states that L-PRP is more effective for female athletes in preventing ACL injuries than P-PRP. Although the authors refer to a paper that provides the incidence date, they do not present any data confirming that L-PRP is more effective than P-PRP. Could you please provide further clarification?
Response: Thank you for this question. This sentence did not refer to the comparison between L-PRP and P-PRP, or between male and female athletes. As mentioned, we intended to introduce the most common choice of PRP in clinical settings. Thus, we suggest that L-PRP is preferred for the treatment of anterior cruciate ligament injury regardless of sex and provides good clinical outcomes in female athletes.
- Line 186. “These phenomena…” Kindly explain the phenomena to which the authors refer.
Response: Thank you for this comment. These phenomena have often been observed in clinical settings and can be monitored by several factors such as serum creatinine kinase levels [Baired et al., J Nutr Metab 2012:960363; 2012]. Regardless of exercise level, skeletal muscle fibers are more or less damaged at the micrometer or submicrometer level. These damages are detected and repaired by a mechanism involving platelet activation and aggregation [Peake et al., J Applied Physiol, 122:559–570;2017]. These are the phenomena referred to in.
- Lines 209-210. Please clarify this sentence: “However, to our perplexity, PRP has rarely been studied either in basic or preclinical studies as a newly developed drug.”
Response: Thank you for this comment. This expression may have caused your misunderstanding. As explained in detail in our previous review article [Kawase and Okuda, Biomed Res Int, 6389157; 2018], PRP was not investigated in the established way that is applied in the process of approval of “new drugs.” New drugs are usually tested in preclinical studies (from in vitro to animal experiments) and then subjected to further testing in clinical trials (from phase I to IV). However, PRP was directly used in clinical settings without any preclinical and clinical studies. We intended to emphasize that this “irregular R&D process” causes the current confusion. We added a reference there.
- Lines 229-234. The data provided here should be in the results, and not be used for the first time in the discussion.
Response: Thank you for this comment. We have moved this part to the Results section and summarized these findings in the Discussion section (#3.2).
- Overall, the discussion is poor, as it does not really go into depth on the results obtained in comparison with other works. No explanations are provided on the differences observed between the two study groups. The authors have several questions that I ask in this review, to deepen the discussion, but there are many more.
Response: Thank you for this comment. In the original version, to avoid speculation-based discussion, we did not expand the discussion too much but simply prepared it. We would like to keep this style in the revised version, but to meet your request below, we added a subsection regarding the limitations. We hope that this addition satisfies you to some extent.
- In the discussion, it is important to include a paragraph explaining the limitations of the study.
Response: Thank you for this comment. We added the subsection regarding limitations in the Discussion section (#3.3).
- Conclusions: Lines 294-296. “These findings suggest that although L-PRP may be less potent in accelerating cell growth, it may be able to control the initial inflammation phase to facilitate tissue regeneration.” Less potent than…? Less potent in athletes than controls? Please, only make conclusions supported by the obtained results.
Response: Thank you for this comment. According to your advice, we deleted the part describing the possible less-growth induction from this sentence.
- Comments on the Quality of English Language
- Moderate editing of English language required.
Response: Thank you for this comment. We have ordered professional Editage English editing service to meet your request.
Round 2
Reviewer 2 Report
L318, Table 1 should be 2.
L33, in athletes should be in female athletes
In abstract, the author should mention the control group are ordinary healthy adults
L274, remember to add space between number and symbol sign, such as age:20−25 should be age: 20 - 25
Edit L228, Owing to change to Due to
L257, it should be Firstly?
L30, add respectively
L81, rewrite
Author Response
Comments and suggestions
- L318, Table 1 should be 2.
Response: Thank you for this suggestion. We corrected it.
- L33, in athletes should be in female athletes
Response: Thank you for this suggestion. We corrected it.
- In abstract, the author should mention the control group are ordinary healthy adults
Response: Thank you for this suggestion. We have modified the sentence above and inserted it into L29.
- L274, remember to add space between number and symbol sign, such as age:20−25 should be age: 20 – 25
Response: Thank you for this suggestion. We corrected this part in line 275. Furthermore, we corrected similar expressions in L291 and 292 and the strength of the correlation (L333, L334, Figures 4, 7, 8).
Comments on the quality of English
- Edit L228, Owing to change to Due to
Response: Thank you for this suggestion. We corrected it.
- L257, it should be Firstly?
Response: Thank you for this suggestion. Both the AI-based software and the editor of English editing suggested “first” there. However, we checked the difference through the internet and corrected “first, second, and third” to “firstly, secondly, and thirdly.”
- L30, add respectively
Response: Thank you for this suggestion. Because body composition indices were determined at once using the weight scale, we thought that “using a bathroom weight scale equipped with an impedance meter” must be better than “respectively.”
- L81, rewrite
Response: Thank you for this suggestion. We rewrote this sentence.
Reviewer 3 Report
- 4.2 Blood collection and preparation of L-PRP. Please, expand the protocol of L-PRP obtention (it is referenced as #24, but it would be more useful and friendly to explain it in the paper) (centrifugation time and speed,…). Please, also confirm if the samples were or no activated)
Response: Thank you for this advice. We added a brief description of the procedure there.
Thank you so much for the clarification. But sorry, but you state at this moment 3 centrifugations, and not two. Please clarify in the manuscript. The platelets (buffy of the second centrifugation) were resuspended in PBS? Not in plasma. Please remember that the plasma (without platelets also contains growth factors and other relevant molecules in regenerative medicine). Great job with the new figure (#9).
- To enhance the value of the paper, it would be worthwhile to include a table showing PLA, WBC, ERI, HTC values for both peripheral blood and L-PRP. Additionally, include the increase in platelets and leukocytes values compared to peripheral blood (either in fold or increase).
Response: Thank you for this advice. We added this data in Table 1.
Thanks you for the table. Please, check the units of the WBC, RBC and PLTs. And additionally, review those of the figures, since they must match.
- Figure 3. It is not clear to this reviewer how the values in the right column panels (b, d, and f) are calculated. Please specify in M&M. I understand that the values in a, c, and e are concentration values, and those in b, d, and f are total values, multiplying the concentration by the volume of L-PRP? Please specify for the sake of clarity (also fig. 5 and others). For example, in the figure 6 it is partially explained “Each level was normalized by PLT count (a, b), WBC count (c, d), or L-PRP preparation (e, f)” but sorry, I don’t understand the normalization of L-PRP (e & f).
Response: Thank you for this question. This is related to your question about “conversion.” In response to your comment below, we have explained that blood cell counts and growth factor levels should be quantified not only by the concentration but also by the total amount injected into the injury site. This is because the concentration can characterize PRP preparation but is not directly linked to the clinical outcomes. The total number of platelets and growth factors is expected to directly influence clinical outcomes. Thus, we believe that these data should be expressed in two ways. The graphs in the right column of Figure 3 represent the total counts of blood cells in the individual L-PRP preparations.
Thank you for your response. All clarified, but I but I believe that for the sake of clarification for the readers, this same or similar explanation should be detailed in materials and methods. The term "conversion" does not seem to me to be very typical in this area, but if it is explained adequately and without doubts, it can be correct.
OK
Author Response
Previous comment: 4.2 Blood collection and preparation of L-PRP. Please, expand the protocol of L-PRP obtention (it is referenced as #24, but it would be more useful and friendly to explain it in the paper) (centrifugation time and speed,…). Please, also confirm if the samples were or no activated)
Previous Response: Thank you for this advice. We added a brief description of the procedure there.
- Thank you so much for the clarification. But sorry, but you state at this moment 3 centrifugations, and not two. Please clarify in the manuscript. The platelets (buffy of the second centrifugation) were resuspended in PBS? Not in plasma. Please remember that the plasma (without platelets also contains growth factors and other relevant molecules in regenerative medicine). Great job with the new figure (#9).
Response: Thank you for this comment and evaluation of Figure 9. We apologize for the error. In this study, we did not prepare platelet suspensions for quantification of platelet ATP and polyphosphate levels. We were confused with a parallel study. Thus, we deleted the procedure for platelet suspension in PBS.
Previous comment: To enhance the value of the paper, it would be worthwhile to include a table showing PLA, WBC, ERI, HTC values for both peripheral blood and L-PRP. Additionally, include the increase in platelets and leukocytes values compared to peripheral blood (either in fold or increase).
Previous Response: Thank you for this advice. We added this data in Table 1.
- Thanks you for the table. Please, check the units of the WBC, RBC and PLTs. And additionally, review those of the figures, since they must match.
Response: Thank you for pointing this out to us. We have corrected these errors.
Previous comment: Figure 3. It is not clear to this reviewer how the values in the right column panels (b, d, and f) are calculated. Please specify in M&M. I understand that the values in a, c, and e are concentration values, and those in b, d, and f are total values, multiplying the concentration by the volume of L-PRP? Please specify for the sake of clarity (also fig. 5 and others). For example, in the figure 6 it is partially explained “Each level was normalized by PLT count (a, b), WBC count (c, d), or L-PRP preparation (e, f)” but sorry, I don’t understand the normalization of L-PRP (e & f).
Previous Response: Thank you for this question. This is related to your question about “conversion.” In response to your comment below, we have explained that blood cell counts and growth factor levels should be quantified not only by the concentration but also by the total amount injected into the injury site. This is because the concentration can characterize PRP preparation but is not directly linked to the clinical outcomes. The total number of platelets and growth factors is expected to directly influence clinical outcomes. Thus, we believe that these data should be expressed in two ways. The graphs in the right column of Figure 3 represent the total counts of blood cells in the individual L-PRP preparations.
- Thank you for your response. All clarified, but I but I believe that for the sake of clarification for the readers, this same or similar explanation should be detailed in materials and methods. The term "conversion" does not seem to me to be very typical in this area, but if it is explained adequately and without doubts, it can be correct.
Response: Thank you for this comment. According to your suggestion, we have added explanations for the data expression in the Materials and Methods section (#4.3 and #4.4). In the field of PRP research, most studies prefer to express growth factor levels as concentrations and seem to ignore the concept of “dose” in PRP treatment. Thus, to call for attention and change the manner of the data expression, we adopted both manners. The term “conversion” does not seem appropriate for some readers in this case. However, we could not determine the best matching term to express this switch. We hope that you accept this term.
Round 3
Reviewer 3 Report
All concerns have been addressed and the manuscript is now ready for publication.
OK